# Die Casting Die Design and Process Optimization of Aluminum Alloy Gearbox Shell

**DOI:** 10.3390/ma14143999

**Published:** 2021-07-16

**Authors:** Mingyu Huang, Qian Zhou, Junyou Wang, Shihua Li

**Affiliations:** 1School of Mechanical Engineering, Nantong University, Nantong 226019, China; huang.my@ntu.edu.cn; 2Jiangsu Wencan Die Casting Co. Ltd., Wuxi 214000, China; junyouok@163.com (J.W.); frank.li@hongbang-nt.com (S.L.)

**Keywords:** aluminum alloy, gearbox housing, mold design, process optimization

## Abstract

Taking an aluminum alloy gearbox of an automobile as an example, according to its structural characteristics, the parting surface was determined, and the initial gating system was designed by using 3D modeling software UG. Based on Magmasoft software, the numerical simulation of the filling and solidification process was carried out to determine the best gating system scheme. The cooling system and core pulling structure were designed, and the parameter design process of the aluminum alloy gearbox shell in the die-casting process was introduced. Aiming at the leakage problem of the gearbox shell in the bench and road test after assembly, the cause was found through numerical simulation and industrial CT analysis, and the problem was solved by adding high-pressure point cooling at the corresponding position of the leakage, and the correctness of the optimization was verified. It provides an effective method for the die-casting production of the transmission housing and the analysis and solution of product defects, which improves the product quality and shortens the production cycle.

## 1. Introduction

The gearbox is a key part of the automobile transmission system, which is equipped with gears for transmission. The internal quality of the gearbox shell is required to be high in strength, air tightness and lightweight [1]. Aluminum alloy has the advantages of low density, high strength, corrosion resistance, wear resistance, good thermal conductivity, easy processing, attractive appearance and so on. It is widely used in automotive, aviation, machinery, communication and other fields [2]. Therefore, most of the aluminum alloy gearbox is used in the industry. The automobile gearbox is generally made of aluminum alloy by die-casting and then processed by mechanical processes. In actual production, defects such as porosity, cold separation and leakage often lead to a low qualified rate of product processing. With the gradual improvement of computer numerical simulation technology, the filling and solidification process of die casting can be simulated to clearly reflect the flow of liquid aluminum in the die casting process so as to predict the internal quality defects of products [3]. It greatly shortens the development cycle and reduces the development cost.

## 2. Mold Design

### 2.1. Product Structure Analysis

Taking the aluminum alloy gearbox housing of an automobile as an example, as shown in Figure 1, the product structure is very complex, and the surface is covered with concave and convex structures such as oil pipeline, reinforcing bars and installation holes, so the mold has a side core-pulling mechanism. The material is AlSi9Cu3. The chemical composition and mechanical properties of the material are shown in Table 1 [4,5]. The shrinkage of the casting is required to be 6‰. The castings have high-density requirements, more oil passage holes, strict leak detection requirements and high porosity requirements. The cross-sample requirements are that the porosity is not higher than 5%, and the pore size is not higher than 3 mm. The average wall thickness of the product is 4 mm, as shown in Figure 2. The wall thickness of local locations (such as the red area) has exceeded 9 mm, and the wall thickness varies greatly. It is difficult to achieve sequential solidification in the later solidification, and it is difficult to die cast.

### 2.2. Establishment of Parting Surface

In the die casting die, the parting surface should generally be selected to ensure that the casting remains at the side of the moving die after die opening to facilitate ejection and should be set at the maximum section of the size profile of the die casting [6]. Because the structure of the product is very complex, the internal structure can not be formed directly, and the parts that are not consistent with the direction of the opening of the moving and fixed die, at this time, the slider needs to be formed. Additionally, the product needs to set a certain demould slope in order to facilitate the later mold opening. The parting surface of the product is shown in Figure 3. Because the opening direction of the upper side, lower side and right side of the gearbox shell are not consistent with the opening direction of the moving and fixed die, in order to open the die smoothly, it is necessary to arrange the upper slider, the slide block and the side slider to form. Figure 3a is the parting line of the sliding block; its maximum size is 313 × 203 mm^2^. Figure 3b is the parting line of the side slider; its maximum size is 262 × 237 mm^2^. Figure 3c is the parting line of the upper slider; its maximum size is 380 × 221 mm^2^. Figure 3d is the parting line of the moving and fixed module; its maximum size is 456 × 359 mm^2^.

### 2.3. Gating System Design

The maximum size of the product is 456 × 381 × 275 mm^3^. The weight of the product is 9.9 kg, and the volume is 3.67 × 106 mm^3^. The blank weight of the product is 10.9 kg, the exhaust of the slag package is about 1.2 kg, the flow channel is about 4.5 kg, and the weight of the whole casting is about 16.6 kg. The inner gate is one of the most important elements in the die-casting process, which directly determines the product quality and production process. It is set in the important or poor flow part of the product as far as possible to ensure a consistent and stable flow pattern in the cavity [7]. Its calculation formula is as follows:(1)$$A_{n}=G∕\rho v_{g}t$$

In Equation (1): *A_n_*—the cross-sectional area of the inner gate, m^2^; *G*—the mass of liquid metal through the inner gate, where 10.9 kg, *ρ*—liquid metal density, 2.7 × 10^3^ kg/m^3^; *ν_g_*—the velocity of the metal liquid at the inner gate, look up the table, take 40 m/s; *t*—filling time, look up the table, take 0.07 s. According to Equation (1), *A_n_* is calculated as 1441 mm^2^.

According to the actual production experience, the area of the cake is about 11 times the area of the inner gate. Considering the location of the inner sprue, the metal liquid should first fill the thick wall and avoid direct impact on the cavity, and the structural features of the gearbox housing, such as the center position. As well as the size of the existing die-casting machine’s clamping system and the distance from the punch to the product center, the side-entry method is used for pouring. At this time, the mold structure is compact, the mold is easy to reach thermal equilibrium, and the gate is easy to remove. The sprue is arranged at the installation position of the side valve plate of the casting, and the liquid aluminum enters the sprue through the split cone, and then it is divided into five horizontal sprues to fill the product at the same time. In order to ensure a certain filling pressure and filling speed, the cross-sectional area of the straight runner should be smaller towards the filling direction of the liquid aluminum, and the cross-sectional area of the horizontal runner towards the inner gate should be smaller. The position of the slag ladle is arranged according to experience, and the position of the slag ladle is adjusted according to the entrapment result of mold flow analysis or the uniform velocity distribution [8,9,10]. The design of the gating system plays an important role in casting quality assurance and mold cost. The fan-shaped pouring system is adopted, which is conducive to ensuring that the distance between liquid aluminum and the cavity is basically equal [11]. UG is a product design and processing Software produced by Siemens PLM Software (Alameda, CA, USA). It was used to design the casting pouring system, as shown in Figure 4.

### 2.4. Cooling System Design

In the die casting mold, the design of a cooling system is beneficial to control the temperature of the mold so that the internal heat reaches a dynamic balance state so as to ensure the quality of the product. The cooling water on the mold is controlled by the cooling water insertion and drainage, with one end of water entering and one end of water exiting so as to reach the balanced state of inlet and outlet. On the one hand, part of the heat will be absorbed in the process of liquid aluminum filling. On the other hand, part of the mechanical energy consumed by liquid aluminum filling will be converted into heat energy. In the early stage of production, a release agent is often sprayed on the parting surface of the mold for the convenience of demoulding, and the volatilization energy of the release agent will take away part of the energy. The cooling system consists of water and oil channels, as well as some high-pressure point cooling mechanisms designed for certain locations to maintain the mold temperature balance inside the mold. Cooling waterways are all over the mold core and are arranged in the parts with high temperature and serious heat. The high-pressure point cooling is set in the area with a long solidification time, and the temperature is cooled separately, and the point cooling core is safe from the product with a certain wall thickness. Figure 5 shows a three-dimensional model of the product gating system with a cooling system.

### 2.5. Design of Core-Pulling Mechanism

For the side hole and concave side areas that are not consistent with the parting direction and are not easy to be formed directly, a core-pulling structure is generally adopted [7]. In the die-casting process, for the gearbox shell, the mold opening sequence is the separation of moving and fixed molds, core pulling first and ejecting later. As shown in Figure 6, the two side holes marked in red in the figure are located on the moving and fixed die, but the direction is not consistent with the direction of the die opening, and there is a certain die drawing angle. Therefore, the two positions are formed with a hydraulic cylinder core-pulling structure. According to the structural characteristics of the product, the two hydraulic cylinder core pulling structures are set on the moving die. As shown in Figure 7, the die-casting mold diagram of the product is shown.

## 3. Selection of Die Casting Process

### 3.1. Selection of Die-Casting Machine

The choice of die casting machine depends on injection energy, chamber capacity, clamping force and mold mounting size. The mold locking force is used to overcome the expansion force during die casting production so as to lock the parting surface of the mold so as to prevent the splash of liquid aluminum [12]. In general, the clamping force of the mold should be greater than the theoretical calculation of the expansion force. Otherwise, the mold can not be locked, and the pressure in the cavity can not be guaranteed, and the liquid aluminum is easy to overflow from the parting surface in the process of filling, resulting in defects such as flying material, which seriously affects the dimensional accuracy of the casting. The expansion force formula can be expressed as follows [7]:

Expanding force when the mold has no eccentricity:(2)P=pA
when there is a core-pulling mechanism, the normal force on the inclined surface of the wedge block is:(3)$$P=p_{1}A_{1}\tan\alpha$$

In Equations (2) and (3), *P* is the bulging force on the parting surface of the die, kN;  p is the injection specific pressure, MPa; *A* is the sum of the projections of casting, gating system and overflow groove on the parting surface, mm^2^; $p_{1}$ is the oblique normal division of the wedge compact block, kN; *A*_1_ projection area of the forming part of the lateral movable core, mm^2^; *α* is the inclination angle of the wedge block.

Calculation of clamping force:(4)$$T=K\times F_{n}$$

In Equation (4), *K* is the safety factor, and 1.2 is taken here, $F_{n}$ is the sum of all the bulging forces.

Check the recommended injection specific pressure of aluminum alloy [7]. For airtight parts, the recommended injection specific pressure is 80–120 MPa, 90 MPa is taken here, and the inclination angle of the slider is 10°. After calculation, the clamping force required should not be less than 31,161.6 kN. According to the calculation results of clamping force, installation size, mold opening stroke, etc., choose a die-casting machine of 3200 T or above and finally choose the die-casting machine model: Buhler 3200 T. In addition, the chamber capacity and installation size of the die-casting machine also meet the requirements. The diameter of the punch used is 140 mm, and the effective length of the pressure chamber is 965 mm. The size of the clamping column on the Buhler 3200 T die casting machine is 1300 × 1000 mm^2^.

### 3.2. Selection of Die Casting Process Parameters

Die casting process design is based on flow, solidification and forming theory. By selecting reasonable die casting process parameters, the die casting production trial run was carried out. The chamber diameter was determined to be 140 mm, and the optimal die casting process parameters, including preheating temperature, pouring temperature and punch speed of the die, were explored. The preheating temperature of the mold is set at 140, 160, 180, 200 and 220 °C, the pouring temperature is set at 650, 670, 680, 690 and 700 °C, and the injection velocity is set at 0.1, 2.5, 3.5, 4.0 and 4.5 m/s [13]. Several orthogonal tests were conducted successively to obtain the optimal process parameters [14,15] through analysis and comparison.

#### 3.2.1. Preheating Temperature of the Mold

The preheating temperature of the mold has an important effect on the quality of the product. Generally, the preheating temperature of the mold is above 180 °C, and it is about 1/3 of the pouring temperature. The thin wall or complex structure of the product can be adjusted appropriately to high preheating temperature, which will improve the quality of the product. If the preheating temperature of the mold is set too low, the shrinkage stress of the casting becomes large, and it is easy to produce cracks. If the preheating temperature of the mold is too high, the preheating time will be increased, the production tempo will be prolonged, and the production efficiency will be reduced. After many trials, the reasonable preheating temperature of the die is controlled at 200 °C for the fixed die and 220 °C for the moving die.

#### 3.2.2. Casting Temperature

The pouring temperature is the average temperature at which the molten metal enters the mold cavity from the pressing chamber, expressed by the temperature value on the holding furnace. The casting temperature has an important effect on the quality of the product. When the temperature is too high, the shrinkage will be large, the gas solubility will be large, and the casting will be prone to crack, coarse grain size and viscous type. Temperatures too low will easily produce insufficient pouring, cold and surface fluidity and other defects. In addition, it is also related to the injection velocity and the composition of the alloy. After many tests, the casting temperature at 670 °C was found to be better.

#### 3.2.3. Injection Velocity

The injection velocity is divided into two stages: slow injection velocity and fast injection velocity. The slow injection stage refers to the moving process in which the punch pushes the aluminum liquid forward until the punch pushes the aluminum liquid in the chamber into the inner sprue, and the fast injection stage refers to the injection speed of the punch when the aluminum liquid is filled with the mold cavity. Fast injection velocity is closely related to mold filling quality. If the injection velocity of the punch is too low, the casting cannot be formed, or the forming quality is poor. Increasing the injection velocity thus improves the fluidity of a molten aluminum liquid, can avoid flow marks, cold separation and other defects. After much practice, the best fast injection speed of the product was 4.5 m/s. The relationship between the injection velocity of the inner sprue and the fast injection velocity is:(5)$$v_{n}A_{n}=v_{k}A_{k}$$
where, *V_n_* is the injection velocity of the inner runner, m/s; *A_n_* is the area of the inner runner, and it can be obtained from Equation (1) that *A_n_* = 1450 mm^2^; *V_k_* is fast injection velocity, 4.5 m/s; *A_k_* is the hole area of the chamber, and the chamber diameter is 140 mm. By calculation, the speed of the inner runner is 48 m/s.

### 3.3. Mold Flow Analysis of Initial Casting Scheme

MAGMAsoft [16] is a casting simulation software made in Germany, which simulates the filling, solidification and other parameters of the casting mold to achieve the overall optimal casting process. The simulation technology of the software makes the complex casting process more digitized and visualized, which is increasingly used in the field of casting [17]. The filling and solidification process of casting were simulated to reflect the internal quality of the product. It is a complex physical change process that changes with time [18,19]. The simulation of the variable shell product was carried out, and the simulation parameters were set in Table 2. The simulation process was infinitely close to the real working conditions, which was conducive to ensuring the authenticity of the casting simulation results. Figure 8 shows a set of pseudo color diagrams used to simulate the filling and solidification process. The different colors on the left side of the casting represent the range of values on the right ribbon. The higher the ribbon, the greater the value.

As shown in Figure 8a, the material flow tracking cloud map shows that the filling of liquid aluminum along the five runners is uniform, orderly, smooth and the filling effect is good. As shown in Figure 8b, when the cavity filling rate is 100%, the casting temperature ranges from 580 to 640 °C, which is higher than the liquid-phase limiting temperature of 570 °C, indicating that the product is not likely to cause defects such as cold isolation in the casting process and the casting quality is relatively good. As can be seen from the analysis results of air entrainment quality in Figure 8c, most areas on the casting surface are blue areas, and a small part of the areas are non-critical areas for setting slag ladles according to their actual positions, which will not affect the products. Therefore, the air entrainment quality of the casting is qualified. As shown in Figure 8d, the solidification cloud diagram of the casting shows that when the solidification time of liquid aluminum is 86.4%, the yellow area marked in the figure has not yet begun to solidify. Therefore, special attention should be paid to this position in the actual leak detection process to observe whether it will produce looseness and lead to leakage. As shown in Figure 8e, the casting filling speed cloud diagram shows that the filling speed of liquid aluminum at the inner gate is 52 m/s, which meets the design requirements. The internal velocity distribution of the product is more uniform, and there is less risk of entrapment.

### 3.4. Die Casting Production

The mold temperature during die casting production is controlled at about 200 °C, and the molded casting is shown in Figure 9. After many tests, the optimal process parameters are as follows: fixed mold temperature is 200 °C, and moving mold temperature is 220 °C; The casting temperature of liquid aluminum was 670 °C. The slow injection velocity is 0.18 m/s, and the fast injection velocity is 4.5 m/s. The injection velocity of the inner runner is 48 m/s. The mold retention time is 30 s. The appearance of the product is less fly-edge, clear outline, no cracks, air holes, cold separation and other defects. After the high and low-pressure leak detection and inspection, all of them were within the range of technical requirements provided by the customer, which proves that the above simulation design is correct and feasible.

## 4. Confirm the Final Casting Process Plan

### 4.1. Problem Proposal

The gearbox housing shall be leak detected with a leak detector before delivery to the customer. According to the standard, the low-pressure chamber should not exceed 12 mL/min, and the high-pressure chamber should not exceed 8 mL/min. High and low pressure are within the design range. However, based on the customer feedback, in the assembly of various parts in the bench test and road test, the low-pressure chamber exceeds the standard of the problem. This shows that from the actual use, due to the vehicle driving, gear transmission and other vibration, and temperature and other working conditions, although it has the strength to meet the requirements, the actual requirements of the leakage is higher than the initial customer proposed standards. According to customer feedback, the leakage point is shown in Figure 10.

### 4.2. Causes of Mold Flow Analysis and Improvement Measures

The solidification time in this position is longer, and the solidification speed is slower than that in other positions, so it is easy to produce defects such as shrinkage cavity and porosity. Therefore, according to the analysis results, the measure of adding high-pressure point cooling inserts at the leakage position was adopted. The high-pressure point cooling inserts were arranged on the side slider and slide block, and the wall thickness at the hole position was cooled on both sides at the same time. The point cooling core has a certain safety wall thickness to the product surface, as shown in Figure 11. After the improvement, the mold flow analysis was carried out again on the casting, the simulation parameters remained unchanged, and the simulation was repeated [20,21]. By observing the solidification cloud diagram, as shown in Figure 12, it was found that when the solidification time of liquid aluminum is 83.34%, the solidification at the hole position marked in Figure 8d was achieved. The simulation results show that it is effective to add a high-pressure point cooling structure to the leakage area.

### 4.3. Industrial CT Test to Verify the Improvement Effect

The internal porosity of the product is detected by industrial CT (full name: computer tomography), and the equipment used is GE’s compact 450 kV CT, which is specially used in the laboratory of nondestructive testing and internal quality testing and is widely used in the foundry field. The equipment has complete functions, and the sample size can reach 500 × 1000 mm^2^. For products with high density requiring high power penetration, it can be equipped with a 450 KV ray tube to achieve better imaging effect [22].

Put the gearbox casing in the CT testing chamber, rotate the product to the final position parallel to the ray source, adjust the voltage and current to see the inside of the workpiece clearly, and set a series of parameters as shown in Table 3. When the product is located in the red display box of the datosx2 acqti software equipped with the CT equipment, the debugging is completed. If the values of voltage and current are appropriate, but the left and right of the red box turn red, a filter should be added until the ratio of Max/Min in the gray value is between 2 and 20, and the Max value is about 11,000, and the Min value is about 1000. At the same time, the voltage in the detection process is not more than 430 kV, and the current is not more than 2400 uA; that is, the power of the equipment is not more than 700 W.

After scanning, manually edit the position of the defect and measure the length of the defect. As shown in Figure 13, each diagram is divided into four parts, the lower right corner is a three-dimensional view of a specific part of the product, and the other three are three views of the position of the product. There is a scale on each picture, which represents the length of the defect icon, and the lower right corner of each picture is the scale of the picture. The larger the zoom ratio, the smaller the length of the icon represented by the same scale on the picture. Through the analysis of the mold flow analysis software Magmasoft, it is concluded that the reason for the leakage is that the internal leakage position of the product is loose and runs through the inner and outer wall of the product. After processing, there are small shrinkage holes in the hole wall, and the inner side of the hole runs through the outer wall of the product. The leakage parts were further confirmed by CT, and the same conclusion was drawn that the leakage location was through, which also verified the accuracy of the mold flow analysis.

One of the major advantages of CT over other X-rays is that it can reflect the location of the problem in three dimensions from multiple angles. As shown in Figure 13, a CT test was carried out on the casting before improvement. The first three drawings showed the defect morphology of the hole position in the fourth drawing from three perspectives, respectively. The picture was marked with a red boxthat indicates the location of the defect and the length of the defect is 10.35 mm. After the improvement, the internal CT of the product is shown in Figure 14, defective location before improvement is marked with green box, which indicates that after the addition of point-cold structure at the leakage location, the internal density at the original defect location is not shown any loose defects, and the analysis and method are effective.

## 5. Conclusions

According to the structural characteristics of the aluminum alloy gearbox shell, the die-casting die design is carried out. The design mainly includes the determination of the parting surface, the design of the gating system, the design of the cooling system, the design of the core pulling structure, etc. The structure of the die-casting die has an important effect on the quality of the casting.

After many tests, the optimal process parameters are as follows: fixed mold temperature is 200 °C, moving mold temperature is 220 °C. The casting temperature of liquid aluminum is 670 °C. The slow injection velocity is 0.18 m/s, and the fast injection velocity is 4.5 m/s. The injection velocity of the inner runner is 48 m/s. When the die time is 30 s, the forming quality of the casting is better, and it meets the technical requirements after the inspection and testing.

Reflecting on the gearbox shell for customers in the assembly after the bench and road tests that have the leaking problem, by simulating the solidification process of leakage location, we found that the location of solidification time is longer, and the solidification is slower relative to other position. It is easy to produce the defects such as shrinkage porosity, and connecting with the industrial CT shows the shrinkage porosity defect. On this basis, a scheme to improve the mold was put forward. High-pressure point cooling inserts were set on the side slider and the slide block, respectively, to accelerate the cooling speed of the position, and the problem was solved.

This optimization scheme provides an effective method for future design, analysis and solution of such problems, which can not only improve product quality and pass rate but also greatly shorten the research and development and production cycle.

## Figures and Tables

**Figure 1 materials-14-03999-f001:**
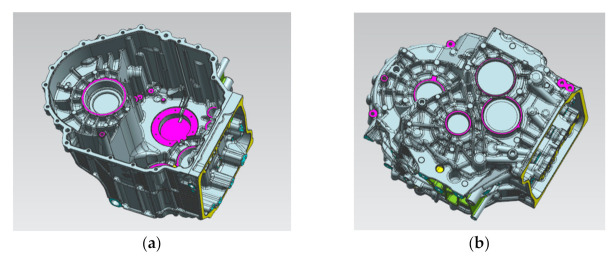
Gearbox structure diagram. (**a**) Bottom view of gearbox, (**b**) Top view of gearbox.

**Figure 2 materials-14-03999-f002:**
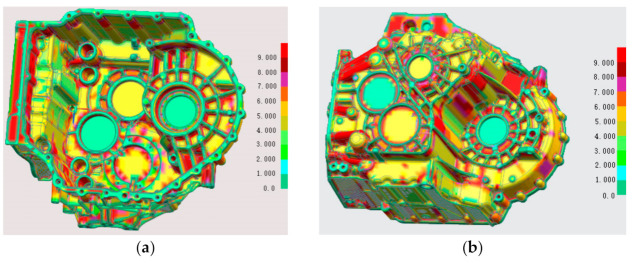
Wall thickness analysis diagram of gearbox. (**a**) Analysis diagram of bottom wall thickness of gearbox, (**b**) Analysis diagram of top wall thickness of gearbox.

**Figure 3 materials-14-03999-f003:**
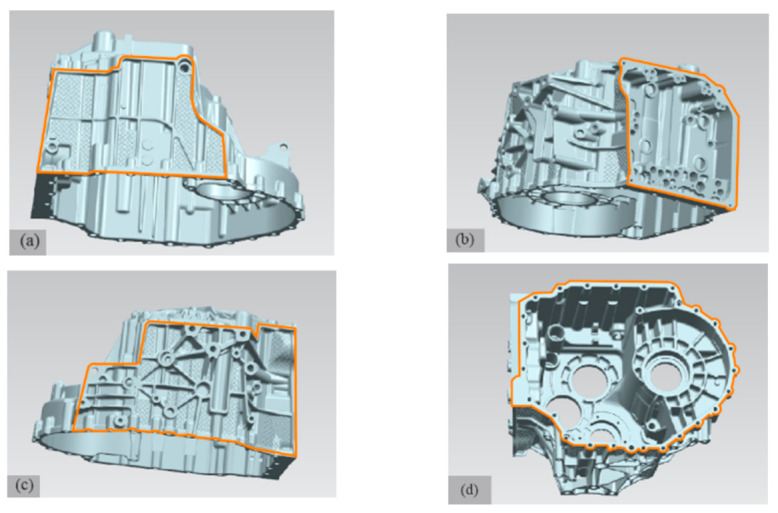
Casting parting diagram: (**a**) parting line of slide block; (**b**) parting line of side slider; (**c**) parting line of the upper slider; (**d**) parting line of moving and fixed module.

**Figure 4 materials-14-03999-f004:**
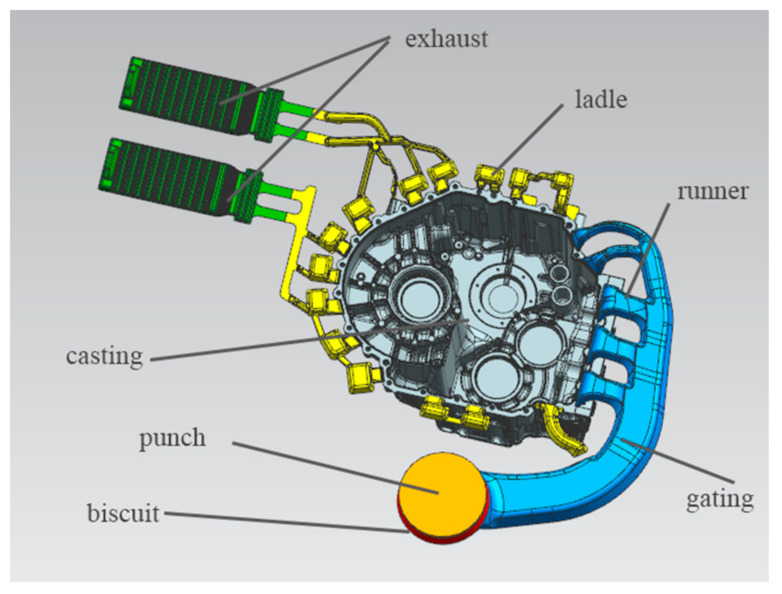
Casting pouring system.

**Figure 5 materials-14-03999-f005:**
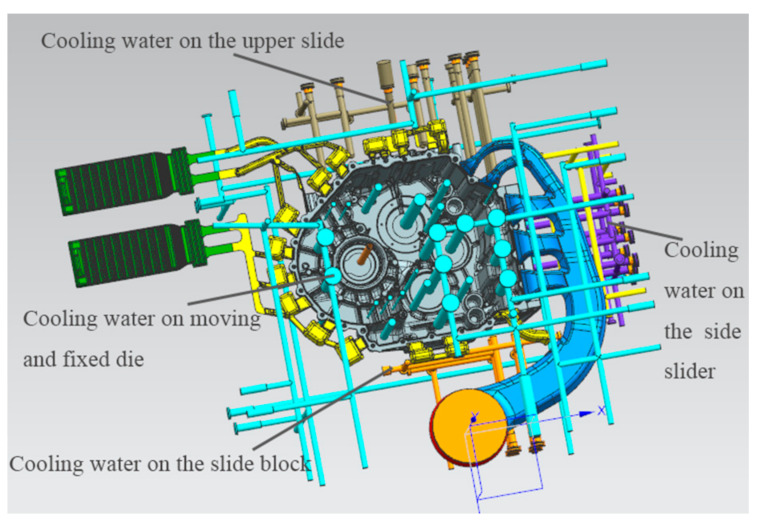
Diagram of product gating system with cooling system.

**Figure 6 materials-14-03999-f006:**
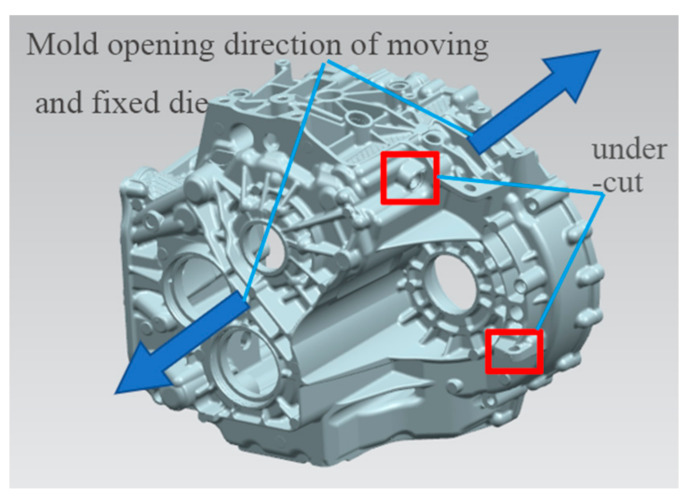
Casting structure diagram.

**Figure 7 materials-14-03999-f007:**
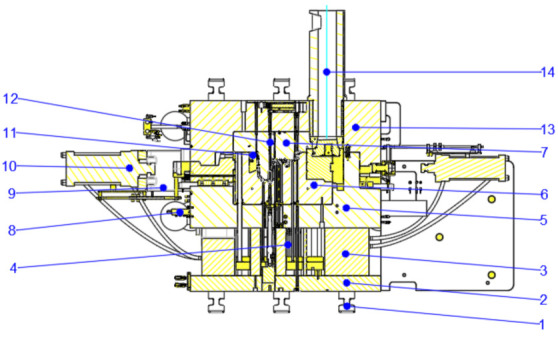
Die Casting Die Diagram: 1—Mold locking column; 2—Base plate; 3—Fixed block; 4—Thimble; 5—Moving mold frame; 6—Moving die core; 7—Fixed die core; 8—Water collector; 9—Cylinder bracket; 10—Oil cylinder; 11—Moving insert; 12—Cooling water pipe; 13—fixed mold frame; 14—melting cup.

**Figure 8 materials-14-03999-f008:**
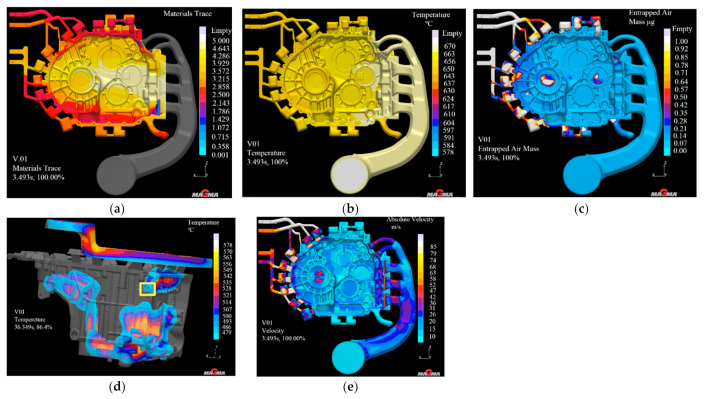
Simulation results of aluminum alloy gearbox housing. (**a**) Material tracking neogram; (**b**) filling temperature neogram; (**c**) entrainment quality neogram; (**d**) solidification neogram; (**e**) filling velocity neogram.

**Figure 9 materials-14-03999-f009:**
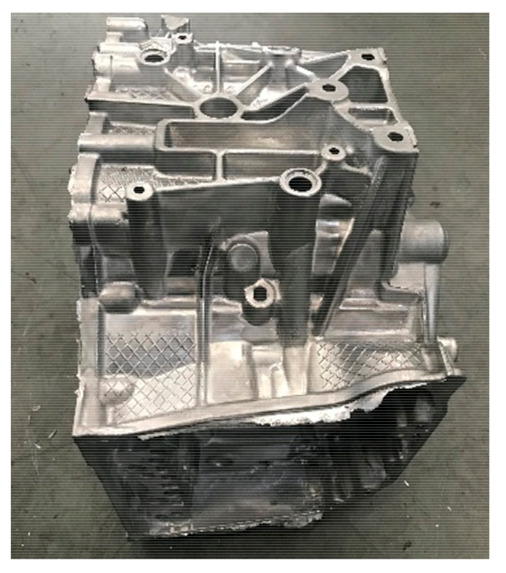
Finished casting drawing.

**Figure 10 materials-14-03999-f010:**
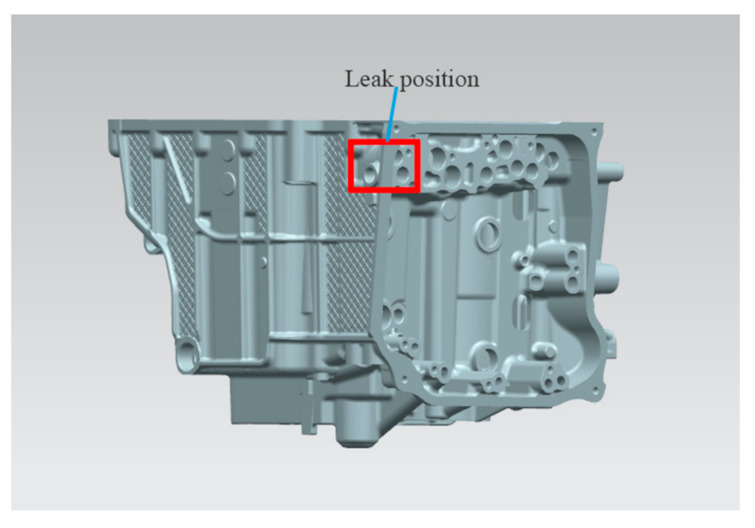
Measured leakage position of the finished gearbox housing.

**Figure 11 materials-14-03999-f011:**
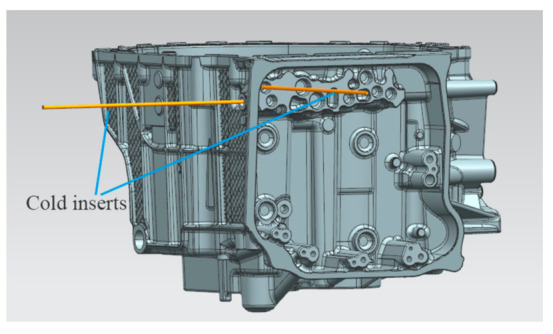
3D model after adding cold inserts.

**Figure 12 materials-14-03999-f012:**
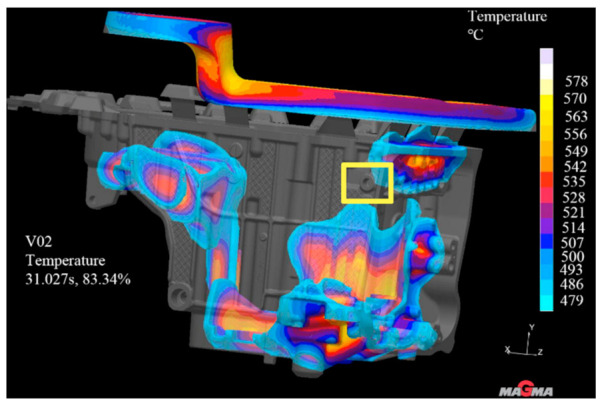
Improved solidification cloud image.

**Figure 13 materials-14-03999-f013:**
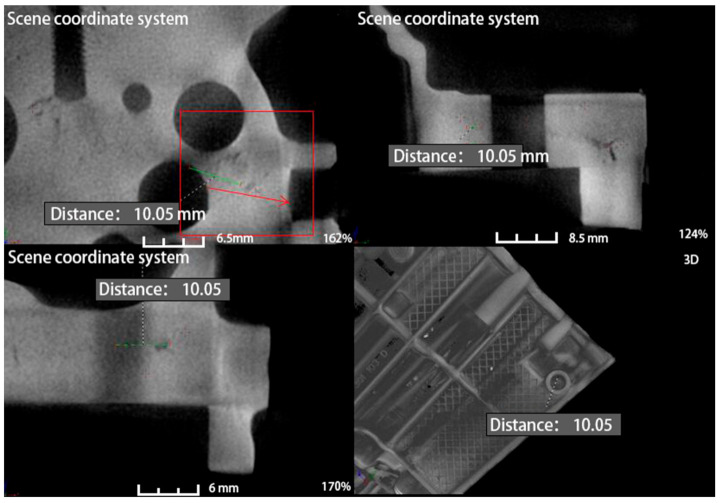
Results of CT measurement at the leakage site before improvement.

**Figure 14 materials-14-03999-f014:**
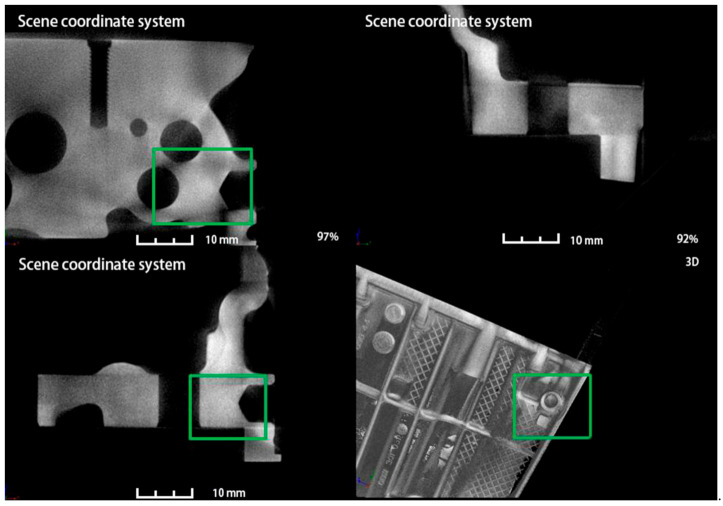
CT measurement results at the leakage site after improvement.

**Table 1 materials-14-03999-t001:** Material Composition and Mechanical Properties.

Chemical Component	Wb/%							Mechanical Properties/≥		
AlSi9Cu3	Al	Si	Cu	Zn	Fe	Mn	Mg	Tensile strength/MPa	Elongation after breaking/%	Yield strength/MPa
	margin	8.0–11.0	2.0–3.5	≤1.2	≤0.8	0.1–0.5	0.1–0.5	240	1	140

**Table 2 materials-14-03999-t002:** List of simulation parameters of variable shell casting.

Die Casting Alloy	Initial Molten Aluminum Temperature/°C	Inner Gate Cross-Sectional Area/(mm^2^)	Pressure Chamber Diameter/(mm)	Casting Pressure/bar	Low Speed/(m/s)	High Speed/(m/s)
AlSi9Cu3	670	1450	ϕ1710	960	0.18	4.5

**Table 3 materials-14-03999-t003:** CT detection parameter setting table.

CT Parameters	Normal Scan	Fast Scan
Timing/(ms)	330	500
Average	1	2
Skip	1	2
Voltage/(KV)	430	380
Current/(uA)	1400	1000
Binning	1 × 1	1 × 1
Sensitivity	1	1

## Data Availability

The data underlying this article will be shared on reasonable request from the corresponding author.

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
