# Peer review of "Die Casting Die Design and Process Optimization of Aluminum Alloy Gearbox Shell"

_materials, 2021, doi:10.3390/ma14143999_

Round 1
Reviewer 1 Report
Please find my comments in the attached pdf.

Author Response
Dear reviewer:
I have replied your comments in the PDF verrsion you sent to me, Please check!

Reviewer 2 Report
referee report
materials-1268157-peer-review-v1
Die Casting Die Design and Process Optimization of Aluminium Alloy Gearbox Shell
Huang Mingyu, Zhou Qian, Wang Junyou
This manuscript reports on the design of a die-cast aluminium gearbox, which is an interesting topic. The design
is done by 3D modelling and numerical simulation of filling and solidification of the metal. The material provided
is interesting, and the topic fits into the scope of Materials.
The manuscript looks at first glance well organized, and it contains 14 figures, 2 tables and 21 references.
A closer inspection reveals several problems which require attention:
# One big problem is that the abstract mentions the names of two software packages, but the main body of the manuscript
does not give any further information. The main problem is that there is no proper section "Experimental details",
which would give the details of the experiments (imaging, CT) nor the model calculations. This must be done including
the proper references to the respective manufacturers.
# The English requires some improvement. Please have a native speaker checking the manuscript.
# There should always be a space between a physical quantity and its unit. Even worse, there are full misprints like
4m/ s.
# Units are not written properly: Pascal is "Pa"
# Formatting is a general problem: If using an index, please write as subscript.
# Equation (4) contains a Chinese character.
# The color scale bars in Fig. 8 are fully unreadable.
# In the CT images, a proper scale bar is required. Please remove all unnecessary (and unreadable) text from the images.
# What does mean [J] and [D] in the reference list?
# Ref. [4]: Science direct is not a journal, but a provider. Volume and page numbers are missing.
# There is no need for "et al." here. Please provide all author names.
Overall, this manuscript contains interesting materials, but also severe deficits which do not allow publication in the
present form.
Author Response
#First of all, I have added information description of two software UG and Magmasoft in the article, including the production company, mainstream functions and so on. Secondly, about the details of the experiment, I have made a supplement in text 4.3, and introduced the experimental equipment, parameters, the company of the equipment, the experimental operation process and matters needing attention in detail. At the same time, a more in-depth description of the graph file of the experimental results is given.
#I have asked native English speakers to check the manuscript and have revised it in the article.
#I've set aside the space between the physical quantity and its units.
#The writing of all MPa units in this article has been corrected.
#The index in the full text has been changed to subscript mode.
#The Chinese characters in Formula 4 have been changed to English.
#I have increased the resolution of the chromatic aberration in Figure 8.
#In the CT image, the scale bar has been added. And includes the magnification ratio. The larger the magnification ratio, the smaller the distance represented by the ruler in the diagram.
#[J]、[D] have been deleted.
#I have added volume and page numbers.
#In the references, I have listed all author names.
Round 2
Reviewer 1 Report
The authors have addressed all of the comments. The quality of this work is very good for publication in this reputed journal.
Author Response
Dear reviewer:
Happy summer vacation ! Some minor spelling and grammar problems have been corrected, please check!
Reviewer 2 Report
The authors have presented a revised version with many corrections and additions. All the changes have clearly improved the quality of the manuscript. Some points still remain:
# If multiplying two sizes, e.g., [mm], the result is to be given squared, i.e., [mm2].
There should always be a space between a physical quantity and its unit.
Please format the reference list uniformly according to the journal's demands. The style of Ref. 2 is quite close, so take this as an example.
Overall, the present manuscript may be published with minor amandements.
Author Response
Dear reviewer:
Happy summer vacation. The modification has been made according to your suggestions. The details are as follows:
First, Multiplying the two dimensions, the units have been changed to mm2; multiplying the three dimensions, the units have been changed to mm3.
Second, a space has been added between all physical quantities and their units.
Third, the references in the paper have been reformatted according to the format of Ref.2.
To sum up, all the changes have been marked in Word with the function of "Track Change". and you can look it up in review mode. Please check!